# Pilot Study: Safety and Performance Validation of an Ingestible Medical Device for Collecting Small Intestinal Liquid in Healthy Volunteers

**DOI:** 10.3390/mps7010015

**Published:** 2024-02-04

**Authors:** Alexandre Tronel, Anne-Sophie Silvent, Elena Buelow, Joris Giai, Corentin Leroy, Marion Proust, Donald Martin, Audrey Le Gouellec, Thomas Soranzo, Nicolas Mathieu

**Affiliations:** 1Pelican Health, 107 rue Aristide Briand, 38600 Fontaine, France; a.tronel@pelican-health.com; 2University Grenoble Alpes, CNRS, UMR 5525, VetAgro Sup, Grenoble INP, CHU Grenoble Alpes, TIMC, 38000 Grenoble, France; elena.buelow@gmail.com (E.B.); jgiai1@chu-grenoble.fr (J.G.);; 3University Grenoble Alpes, Inserm, CHU Grenoble Alpes, CIC, 38000 Grenoble, France; silvents@univ-grenoble-alpes.fr (A.-S.S.); cleroy@chu-grenoble.fr (C.L.); mproust@chu-grenoble.fr (M.P.); 4Service de Biochimie Biologie Moléculaire Toxicologie Environnementale, UM Biochimie des Enzymes et des Protéines, Institut de Biologie et Pathologie, CHU Grenoble-Alpes, 38000 Grenoble, France; 5Plateforme de Métabolomique GEMELI-GExiM, Institut de Biologie et Pathologie, CHU Grenoble-Alpes, 38000 Grenoble, France; 6Department of Hepato-Gastroenterology and Digestive Oncology, Grenoble Alpes University Hospital, 38000 Grenoble, France

**Keywords:** medical device, ingestible, pilot study, small intestine, microbiome, multi-omics, heathy volunteers

## Abstract

The connection between imbalances in the human gut microbiota, known as dysbiosis, and various diseases has been well established. Current techniques for sampling the small intestine are both invasive for patients and costly for healthcare facilities. Most studies on human gut microbiome are conducted using faecal samples, which do not accurately represent the microbiome in the upper intestinal tract. A pilot clinical investigation, registered as NCT05477069 and sponsored by the Grenoble Alpes University Hospital, is currently underway to evaluate a novel ingestible medical device (MD) designed for collecting small intestinal liquids by Pelican Health. This study is interventional and monocentric, involving 15 healthy volunteers. The primary objective of the study is to establish the safety and the performance of the MD when used on healthy volunteers. Secondary objectives include assessing the device’s performance and demonstrating the difference between the retrieved sample from the MD and the corresponding faecal sample. Multi-omics analysis will be performed, including metagenomics, metabolomics, and culturomics. We anticipate that the MD will prove to be safe without any reported adverse effects, and we collected samples suitable for the proposed omics analyses in order to demonstrate the functionality of the MD and the clinical potential of the intestinal content.

## 1. Introduction

The human gastrointestinal tract (GIT) consists of various segments, each characterised by distinct environmental conditions. These environmental conditions play a direct role in shaping the microbiome of the GIT. The gut microbiome, a complex ecosystem, encompasses communities of microorganisms, including bacteria, fungi, and archaea, along with their biological components like metabolites, proteins, or free nucleic acids [1]. The human gut microbiome exhibits a stratified distribution from the duodenum to the colon, characterised by a progressive increase in bacterial density and diversity (from 10^3^ CFU/mL to 10^11^ CFU/mL) along this continuum [2]. It is well established that the human gut microbiome plays a crucial role in preserving the host’s health, directly influencing metabolic and immune homeostasis. Gut microbiome dysbiosis is characterised by a decrease in microbial diversity, the absence of beneficial microorganisms, and the potential presence of pathogenic microbes, and it can lead to alterations in the composition of gut microbiota-derived metabolites, thereby contributing to the development of various pathologies [3,4]. Numerous studies have established connections between the intestinal microbiome and conditions such as metabolic diseases, inflammatory bowel diseases, cancers, and neurological disorders [5,6,7,8,9,10]. Gut microbes exert a direct or indirect influence on the host’s physiology by regulating various hormones, neurotransmitters, or metabolites [11]. Certain metabolites, like Short Chain Fatty Acids (SCFA), are known to have direct impacts on gut physiology, exhibiting anti-inflammatory effects, maintaining the integrity of the gut barrier, and serving as the primary energy source for colonocytes [12,13,14]. These metabolites are either directly produced by gut microbes, from diet, from host metabolite such as amino-acids conjugated bile acids, or de novo [4].

The human gut microbiome has been mainly characterised based on the analysis of faecal samples through the cultivation of gut microbes and DNA based next-generation sequencing [15]. However, these samples do not represent the intestinal microbiome [16]. The faecal microbiome is dominated by four phyla: *Bacillota*, *Bacteroidota*, *Actinomycetota*, and *Pseudomonadota* [17] and has been extensively studied. The upper intestinal tract (the small intestine) is less characterised since its accessibility is limited, and it is known to be less diverse than the faecal microbiome and dominated by the genera *Lactobacillus*, *Clostridium*, *Streptococcus*, *Staphylococcus*, *Veillonella*, and *Bacteroides* [18,19,20,21,22]. The gut microbiome’s composition of the small intestine is strongly influenced by the short transit time of ingested food, the pH and oxygen gradients, as well as enzymes and bile acids produced by the host [14,20]. Studies that aimed to characterise the small intestine employed different collection and analytical methods, thus making comparisons between studies difficult.

Microbiome sampling plays a pivotal role in clinical investigations, aiding in the understanding of gut microbiome dynamics and their implications for human health. To date, various methods for gut microbiome sampling are available, but most of them are invasive for patients and costly for health care facilities. Endoscopic aspiration, while prone to contamination [23], is essential for diagnosing conditions like small intestinal bacterial overgrowth (SIBO) [24]. Luminal brushing offers precision and minimizes contamination but may require time and effort [25]. Surgery, which is non-susceptible to contamination, theoretically provides the most representative samples, yet necessitates preoperative preparations that can disrupt the microbiome [26]. In vivo models, such as ileostomy effluent, offer convenience and insight into diet-microbiota interactions but may not fully replicate the microbiome of individuals with normal anatomy [27]. While these approaches have made significant contributions to clinical research, the invasive nature of surgery and the difficulties associated with obtaining samples from healthy controls constrain their broad utilization. The choice of sampling method in clinical settings should consider the specific research goals and practical constraints while aiming to strike a balance between accuracy and invasiveness. Tang et al. have detailed different approaches currently used for sampling the gut microbiome such as endoscopy or biopsy [28]. To date, some research teams have developed non-invasive methods to sense and collect intestinal liquid by using smart capsules [29].

### 1.1. Rationale

The gastrointestinal tract is a complex ensemble of organs performing a variety of crucial functions in the digestion, absorption, and processing of food and nutrients, as well as in the regulation of the immune system and overall health [30]. Exploring the gastrointestinal tract is essential to gain a deeper understanding of its intricate workings and the profound impact it has on human health, paving the way for more effective diagnostics, treatments, and preventive measures for a wide range of gastrointestinal and systemic diseases [31]. The development of advanced instruments, like endoscopes, in the 19th and 20th centuries, allowed for the direct visualization and exploration of the gastrointestinal tract. Endoscopy, with its various forms such as gastroscopy and colonoscopy, revolutionized the field of gastrointestinal medicine and diagnostics. Even though the technique is minimally invasive, it still requires a medical environment and a clinician, and it is not well accepted by patients due to the great discomfort it causes. Moreover, endoscopy does not allow us to reach parts of the small intestine. To circumvent some of these disadvantages, ingestible devices were developed. Besides the exception of the “Heidelberg capsule” in 1957 for pH measurements [32], smart wireless capsules have been entering the market since 2001, with the PillCam (Given Imaging) imaging video capsule for imaging the gastrointestinal tract [33]. Since then, various other ingestible medical devices have been developed for different purposes, including smart pills for drug delivery and monitoring [34,35]. These devices have significantly advanced medical diagnostics and treatments for gastrointestinal and other related health issues [36,37]. To date, wireless capsule endoscopy is the gold standard for the diagnosis of different small intestinal diseases such as Crohn disease, angiodysplasia, or polyposis [38]. Because a lot of biophysical processes happen in the intestine and since it hosts billions of microorganisms known as the microbiota, sampling the intestinal content has become of great interest [31,39]. Following the trend of ingestible medical devices, multiple devices have been developed to perform sampling, including the Intellicap^®^ [35]. This smart pill was a sophisticated programmable capsule that obtained a CE marking. Since then, other projects have emerged as reviewed elsewhere (Rehan et al., [29]). To sample the human microbiome in a non-invasive way and in vivo, Pelican Health has developed an ingestible device that contains no electronics and that has the unique ability to collect three samples at a time. This article details the clinical protocol used to evaluate the Pelican Health medical device on healthy volunteers for the first time.

### 1.2. Medical Device Description

The Medical Device (MD) is developed to specifically target the small intestine, and it is designed to be easily recovered from faeces. It is a porous polymer protected by two walls. The outermost layer is gastro-resistant, in the form of a size 00 gastro-resistant pharmaceutical capsule. This capsule is produced by Lonza Capsules & Health Ingredients (Colmar, France) and targets the distal parts of the small intestine [40]. This enables the capsule to pass through the stomach without being damaged. The outer wall (capsule) then degrades once it reaches the distal jejunum/proximal ileum. The MD contains tree modules, allowing for the multi-sampling of the small intestinal liquid. These modules are radio-opaque, allowing for the location of the MD in the GIT, if needed, using an X-ray or a scanner. The device is designed in compliance with medical device design standards, using biocompatible, certified materials and has no electronic materials or energy sources, thus limiting production risks and costs. 

### 1.3. Aims and Objectives

The primary objective of this clinical study is to demonstrate the safety and performance of the medical device. This objective is, thus, composite and will be achieved if both the safety and the performance of the MD are observed for at least 12 out of 15 volunteers. 

Each volunteer will ingest only one pill (the MD) containing tree modules.

The secondary objectives of this study are as follows: (1) to verify the safety of use of the MD for the collection of intestinal content, (2) to verify that the MD is transiting in the human intestinal tract, (3) to evaluate the capture of intestinal liquid by the MD, (4) to verify that the MD opened in the intestine, (5) to verify the state of the MD after collection, (6) to evaluate the sample’s functional potential and identify biomarkers (microorganisms, metabolites) via metagenomics and metabolomics, (7) to estimate the volunteer’s experiences about the MD ingestion and its use, and (8) to obtain the sample’s functional potential and identify biomarkers via culturomics for two subjects.

### 1.4. Primary Outcomes

Safety will be assessed in a binary manner as the absence of the occurrence of adverse events of grade ≥3 with a proven or reasonably conceivable causal relationship with the MD or its related procedure. Grades will be determined using the common terminology criteria for adverse events (CTCAE) classification version 5 (or 6 if available) [41]. 

Performance will be considered satisfactory if at least one of the three modules is analysable. A module will be considered analysable if (1) it is found in the faeces within a 96-h period, (2) the recovered volume of its contents is ≥25 µL, and (3) its visual condition is satisfactory (no faecal contamination).

The criteria to evaluate the secondary objectives are as follows: (1) the number and grade of adverse events notified (including response to follow-up volunteer-diaries, number of unscheduled medical visits); (2) the number of modules recovered per volunteer; (3) the volume recovered from each module (recovered volume must be ≥25 μL); (4) the pH of the recovered MD contents (a measured pH ≥5 will prove that the MD did not sample stomachal liquid where the pH is lower [42]); (5) the visual assessment of the module evaluating whether it is intact and has no visible cracks or damages that could lead to faecal contamination; (6) metabolomic and metagenomic analyses of the module contents; (7) the rating of the device (devices “acceptability”) based on the questionnaire and the end-of-participation visit with the principal investigator; and (8) the results of the culturomic analysis of the module contents, on up to two volunteers.

## 2. Experimental Design

This investigation is an interventional and monocentric study conducted at the Grenoble Alpes University Hospital on 15 healthy volunteers. 

### 2.1. Inclusion Criteria

To be eligible for inclusion in this study, participants must meet the following criteria: (1) be aged between 18 and 65 years; (2) have a BMI higher than 20 and lower than 30; (3) be affiliated to a social security system; (4) be able to understand the study and all the associate instructions; (5) must have a transit time evaluated from 1 to 3 faeces per day (evaluated during the inclusion visit); (6) accept to maintain the same lifestyle (sport and diet) during the study duration; (7) stay in the Isère (French department) during the faeces collection time (around 96 h); (8) must have signed the study’s informed consent and collection consent. 

### 2.2. Exclusion Criteria

Exclusion criteria for this study includes the following: (1) subjects taking medication (except oral contraception) and/or probiotics and related products, (2) subjects with a known food allergy, (3) subjects who have had an intestinal obstruction, (4) subjects with inflammatory disease of the digestive tract and/or history of digestive surgery. More exclusion criteria are described in Appendix A.

### 2.3. Participants and Recruitment

Both male and female volunteers who live in Grenoble and its surrounding areas will be recruited. The participant recruitment started in October 2022 and will be completed in December 2023.

### 2.4. Data Collection

Pre-analytical assessments will be performed inside the Biology and Pathology Institute of the Grenoble Alpes University Hospital. Metabolomic, metagenomic, and culturomics analyses of the module contents will be performed as follows: the metagenomics analysis will be performed by ADM biopolis (Valencia, Spain), the metabolomics analysis by GExiM (Grenoble, France), and the culturomics analysis by the TIMC lab (Grenoble, France).

### 2.5. Clinical Study Protocol

During the screening visit, which takes place about a week before the inclusion visit at the hepato-gastroenterology department, the investigator informs the volunteer and answers any questions about the objective, the nature of the constraints, the foreseeable risks, and the expected benefits of the study. The investigator also explains the volunteer’s rights in a clinical trial and outlines the eligibility criteria. 

After receiving this information, volunteers are given at least 24 h to consider their decision to participate in the study or not. If, after the reflection period, the volunteer does not wish to give consent, he or she simply informs the research team by telephone or e-mail.

Once included in the study, on the appointed day, the volunteer comes to the hospital in the morning after a 10 h fasting to swallow the MD with 150 mL of water under the supervision of the study gastroenterologist. For women of childbearing potential, a pregnancy test is performed to exclude a pregnancy. After ingesting the MD, the volunteer must then wait 2 h before drinking and 4 h before eating. Consequently, the volunteer collects each faeces until the clinical study assistant calls him to stop the collection once all modules have been collected. For the faeces collection, the volunteer uses a special device named Fecotainer that can be placed directly in the toilet bowl. The collected sample must only contain faecal matter (no urine or blood) to be analysed. After defecation, the fresh sample is directly placed by the volunteer in a refrigerated place, and a biological transporter collects the sample from volunteer’s house to the hospital 24/7. Depending on the delivery hour at the hospital, the sample is directly prepared (from 8 h to 18 h) or placed at +4 °C (from 18 h to 8 h). Samples must be prepared in the next 24 h after collection and must be stored at +4 °C. The volunteer keeps track of their nutrition and sensation (presence or absence of pain, etc.) during study duration on a daily basis. 

The sample preparation consists first in screening for the module in the collected faecal matter. If no module is detected, the faeces collection continues. If at least one, or all modules, are detected in the collected faecal sample, the modules are recovered. The collected modules are then washed and assessed (e.g., observed leakage), weighted (to estimate the intestinal liquid volume sampled), and the content is recovered by centrifugation, leading to a sample that is divided into a pellet (bacterial content) and supernatant (intestinal metabolites). The supernatant is transferred into a new Eppendorf tube for metabolomic analysis, and its pH is measured using a pH paper indicator. The pellet is used for metagenomic analysis. For each volunteer, one module is used for multi-omics analysis, and the remaining ones (if any) are stored in the biological collection in case of a complementary analysis and as backup. If a module is contaminated by faecal matter due to leakage, the back-up module is used for the multi-omics analysis. Faecal samples are also collected from healthy volunteers and will be analysed by the same approaches as the intestinal content collected with the medical device. All samples are stored at −80 °C until analysis. 

For two volunteers, culturomics analysis will be performed if at least two out of the three modules per healthy volunteer are recovered and contain more than 50 µL, displaying no faecal contamination. The first retrieved module will be used for metagenomics and metabolomics analysis, and the second retrieved module will be used for culturomics analysis and will be directly placed in an anaerobic bag to preserve strictly anaerobic bacteria. The samples will be rapidly sent to the culturomics platform at +4 °C. In case the third module is contamination-free, it will serve as a back-up for analysis. 

The design of the study is illustrated in Figure 1.

### 2.6. Multi-Omics Analysis

Three meta-omics analysis will be performed during this clinical study (Figure 2). All samples will be analysed simultaneously and at the end of the study to avoid batch-related analysis biases, expect for culturomics. 

Metagenomics will be performed on one (small intestinal content (bacterial pellet obtained by centrifugation) + corresponding faeces) sample for each of the 15 healthy volunteers using 16S rRNA gene sequencing to determine the microbial composition of each sample. DNA from each sample will be extracted using Qiagen powerfecal pro Kits. Each sample will be amplified through PCR using a specific combination of primers specially designed and adapted for massive sequencing that allows us to capture the hypervariable region V3-V4 of the bacterial 16s rRNA [43]. To be able to construct the bacterial profile, each PCR product will be individually labelled. Each labelled product will pass through a primer-dimer removal protocol in order to allow us to increase the sequencing throughput. Once the PCR products are cleaned, they will be equimolarly pooled. The absolute abundance of 16S rDNA copy numbers representing bacterial biomass for each sample will be determined using qPCR.

Metabolomics will be performed on one (small intestinal content (liquid phase obtained by centrifugation) + corresponding faeces) sample for each of the 15 healthy volunteers. Small intestinal content samples collected by the MD and faecal matter will be analysed using liquid chromatography (Vanquish Flex, Thermo Fisher Scientific, Waltham, MA, USA) coupled with tandem Q Exactive Plus Orbitrap mass spectrometer (Thermo Fisher Scientific, Waltham, MA, USA) (LC-MS/MS). Untargeted and semi-targeted metabolomics will be performed in both positive and negative ionisation modes to characterise the metabolome of each sample. The main goal is to verify whether the volume collected by one module is sufficient to measure its metabolic fingerprint and to look for any differences with faecal matter. Principal component analysis on the metabolic fingerprints will allow us to conclude on this matrix difference. The semi-targeted approach will allow us to compare the relative amounts of approximately 45 preselected metabolites of interest between the small intestinal content and faeces.

Culturomics will be performed on two samples collected by the MD from two different healthy volunteers. For this, the small intestinal liquid recovered from the module will be diluted ten times in anaerobic phosphate-buffered saline (PBS) before being directly inoculated on Columbia +5% blood sheep agar or Yeast Casitone Fatty Acids (YCFA-modified agar in serial dilutions. Alternatively, the small intestinal liquid will be mixed with two liquid media (blood bottle enriched with rumen fluid and sheep blood, and YCFA medium enriched with rumen fluid and sheep blood) under anaerobic conditions. The same media and dilution series will also be cultivated under aerobic conditions. The incubation of cultures will be performed for a total of 10 days, with subcultures at 3 h, 6 h, 9 h, 24 h, 72 h, 7 days, and 10 days from the initially incubated liquid media (blood bottle enriched with rumen fluid and sheep blood, and YCFA medium enriched with rumen fluid and sheep blood). MALDI-TOF mass spectrometry will be carried out to identify bacteria after colony isolation. Species identified through MALDI-TOF will be confirmed via the sequencing of their 16S DNA genes. Isolates that will not be identified through MALDI-TOF will be analysed using full-length 16S rRNA gene sequencing. Strains will be banked and stored at −80 °C.

### 2.7. Sample Size

The number of subjects to be included is set at 15 healthy volunteers to meet practical feasibility requirements. Any patient whose data are deemed unusable, notably due to sample transport lasting >4 h, will be replaced to obtain 15 usable samples, up to a limit of 20 patients in total.

### 2.8. Statistical Analysis

Categorical variables will be presented using counts and frequencies, while continuous variables will be presented using mean, standard deviation, median, and interquartile range. Statistical significance will be declared when *p* < 0.05, without adjustment for multiple testing. Any comparisons between groups will be made using non-parametric tests, given the small sample size: Fisher’s tests for qualitative variables and Wilcoxon’s tests for quantitative data. Analyses will be performed using R (version ≥4.2). The normality of continuous variables will be assessed graphically. No replacements of missing data will be performed.

The main objective of the study will be achieved if the safety and performance sub-criteria are met for at least 12 out of the 15 volunteers. All secondary objectives will be analysed using descriptive tools except multi-omics analyses. 

A security interim analysis will be performed after the inclusion of the 3rd volunteer. This will focus strictly on the safety component of the primary outcome. No adjustments for alpha risk inflation are planned.

### 2.9. Ethical Approval and Registration

This study has been granted ethical approval by the Personal Protection Committee (23 February 2022 and 9 March 2023) and by the French National Agency for the Safety of Medicines and Health Products (ANSM) (2 June 2022 and 20 March 2023), and it has formally been registered as a study (NCT05477069). When the data have been evaluated, participants who request to see a summary of the study results will be given that information. Publication in a peer-reviewed journal and presentation at both national and global scientific conferences will be the primary means of disseminating the study findings from this study.

## 3. Expected Results

The small intestinal microbiome remains inadequately investigated, with current methods of sample collection posing invasiveness concerns for patients. Recognizing the significance of exploring this microbial community, we advocate for the development of a medical device that addresses the drawbacks associated with the invasive sampling of the gut microbiota. Such a device would pave the way for a more patient-friendly approach, facilitating the exploration of the small intestinal microbiome. Moreover, the untapped potential of metabolites exclusive to the small intestine, such as conjugated primary bile acids [44], and bacterial strains specific to this region [18,20,23], further emphasizes the necessity for improved sampling techniques. Beyond alleviating patient discomfort, the utilization of this medical device holds potential for clinicians to uncover novel biomarkers—whether bacteria or metabolites—establishing crucial links to various diseases. In essence, the innovation in sample collection could not only enhance our understanding of the small intestinal microbiome but also open new avenues for diagnostic and therapeutic advancements in medical practice.

We predict that the medical device will be fully safe for the 15 volunteers and that no side-effects will be noticed. Moreover, we expect to achieve small intestinal liquid collection as well as to recover all modules from the faecal matter of the included volunteers. We anticipate that the volunteers’ feelings about the ingestion and use of the medical device will be positive. To perform multi-omics analysis, we presume that the volume collected by the modules will be sufficient and that samples will be analysable to determine the functional potential of each sample. We expect to find differences in profiles depending on sample origins (faecal matter vs. small intestinal liquid). The bacterial composition is expected to be more diverse and richer in faecal samples compared to small intestinal samples according to the literature [23]. In a previous clinical investigation performed with a collection module, Shalon et al. showed distinct microbial communities depending on the collection site [44]. They notably found a higher abundance of *Pseudomonadota* in small intestinal samples than in faeces and a lower alpha diversity relative to the intra-individual diversity in intestinal samples compared to faeces. We also expect to observe differences in the metabolome. In the same clinical study, Shalon et al. analysed the bile acid profiles along the intestinal tract. As a result, they obtained divers bile acid compositions with more primary bile acids (Cholic acid and Chenodeoxycholic acid) in the small intestinal samples, and faeces were dominated by secondary bile acids [44]. The culturomics analysis will allow us to isolate microbes and potentially to identify new bacteria. These bacteria could possibly be new probiotics [45].

Pelican Health is proactively addressing the burden of faeces collection imposed on volunteers in the current study by ensuring that pick-ups align with the volunteers’ way of life in this study. Currently, we are developing approaches to facilitate both the recovery of faeces and the medical device recovery from faeces, thereby lowering additional costs and patient burden. 

Sample contamination is a common issue frequently observed during the employment of other non-invasive sampling tools [46], and even during catheter aspiration [18]. However, the design and technical constraints incorporated into the development of the Pelican Health sampling device are specifically intended to limit these situations. 

The perspectives of this work are numerous. First, delving into the study of the unknown in vivo microbiome through this clinical investigation will demonstrate the safety and efficacy of employing Pelican Health’s medical device for sampling the small intestinal liquid. This marks a crucial advancement in research, as the comparison between small intestinal liquid and faecal samples presents an opportunity to reveal functional differences that could unlock the mysteries of this current “black box” in microbiome research.

Secondly, the potential to extend this study to clinical trials involving diseased patients holds promise for identifying specific biomarkers linked to various diseases. By leveraging the Pelican Health medical device in a clinical context, researchers can explore how the small intestinal microbiome evolves in the presence of diseases, providing insights that could revolutionize diagnostic approaches. This will not only enhance our understanding of pathologies but will also create a pathway for the development of targeted diagnostic markers.

In a third step, the findings from this research could pave the way for treatment potentials. By identifying markers associated with a healthy gut, potentially through comparisons with samples from volunteers, the research may contribute to the development of therapeutic interventions. Understanding the nuances of a healthy small intestinal microbiome can guide efforts to restore microbial balance in patients with dysbiosis or other related conditions. In summary, this multifaceted approach not only deepens our understanding of the small intestinal microbiome but also holds immense potential for clinical applications, from diagnostics to treatment strategies.

## Figures and Tables

**Figure 1 mps-07-00015-f001:**
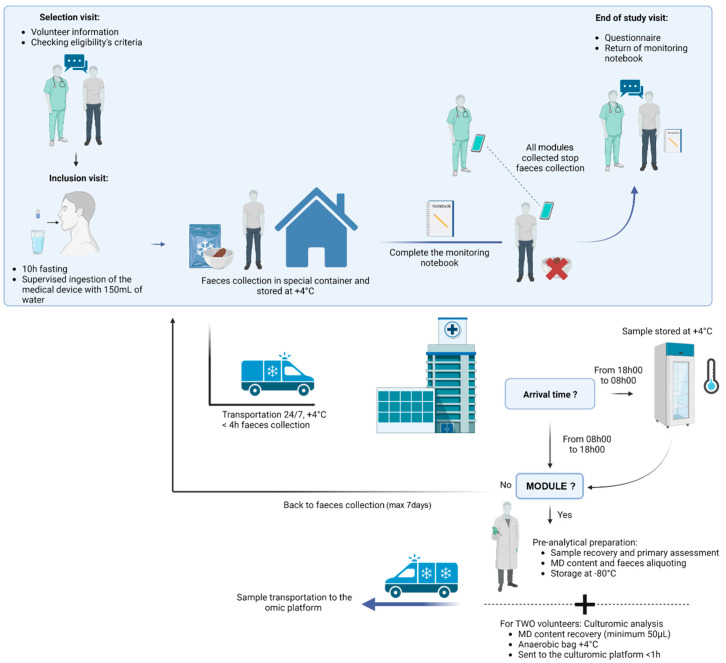
Clinical investigation overview for any given healthy volunteer.

**Figure 2 mps-07-00015-f002:**
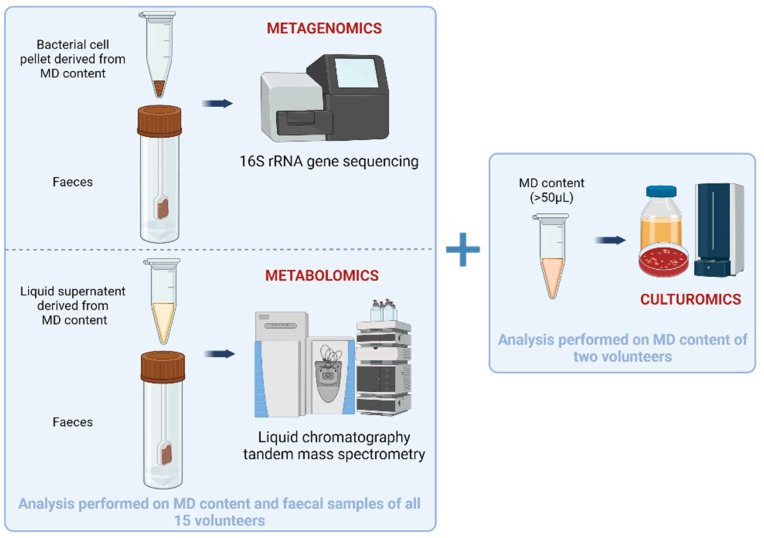
Multi-omics analysis performed for the clinical investigation.

## Data Availability

No data available yet. Obtained data will be made available upon completion of the study.

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
