# Peer review of "Pilot Study: Safety and Performance Validation of an Ingestible Medical Device for Collecting Small Intestinal Liquid in Healthy Volunteers"

_mps, 2024, doi:10.3390/mps7010015_

Round 1

Reviewer 1 Report

Comments and Suggestions for Authors

The peer-reviewed manuscript is well structured and contains sufficient information to understand the relevance and novelty of the experimental research.

I had several questions about the experimental part of the study.

1. It is planned to receive 3 capsules from volunteers; each capsule should contain only the liquid fraction. It is assumed that the pellets obtained by centrifugation of this fraction contain bacterial cells; therefore, it is planned to be used for multi-omics analysis. In this analysis, the authors plan to conduct an analysis based on amplicon metasequencing. However, this is a fairly limited analysis that will provide a list of species and information for predictive conclusions. Currently, there are methods that allow one to analyze the composition of low-abundance microbial communities. Deep metagenomics has significant advantages: it is possible to obtain (1) a list of the predominant species that implement (2) a specific metabolic profile, which can be controlled by (3) the virome. Moreover, the authors must understand that the capsule with its contents will be in conditions favorable for further growth and reproduction of bacteria for a certain time. PCR methods are very sensitive and can give a false impression of the structure of such communities.

2. To what extent is the 25 µl of liquid fraction available in the culture capsule intended to be used? Given that the cultivation will be carried out with dilution, the entire remainder of this part of the liquid fraction could be used for an alternative metagenomic study, and this is where it makes sense to take amplicon libraries. Obviously, the volume of starting material will be less, but using PCR you can be guaranteed to obtain the target product.

Author Response

Answer to reviewer 1:

We thank the reviewer 1 for his/her questions and suggestions. The application of amplicon metabarcoding instead of other methods (such as shotgun sequencing) is justified since we first intend to evaluate the efficacy and performance of our medical device and information on sample recovery, quantity and quality so far is limited and still must be determined. Amplicon based microbiota characterization is a standard method for the evaluation of the richness and diversity of the gut microbiota and still more accessible on a practical and financial level for most labs and industries.   

In future clinical investigations, we plan to perform other metagenomics analysis to increase the panel of applicable analysis. We thank the reviewer for this very interesting suggestion.

Concerning the culturomics part, a minimal volume of 50µL of intestinal content is required for analysis. In case we achieve to sample the required volume, the entire sample is then used for direct inoculation or for pre-culture cultivation. Hence, it is unfortunately not possible to include additional analysis such as amplicon sequencing or metagenomics. We made that clearer in the main text (lines 242-248).

Reviewer 2 Report

Comments and Suggestions for Authors

The manuscript intriguingly addresses a significant gap in current research methodologies concerning the evaluation of gut microbiota and metabolites, which predominantly rely on stool samples. This reliance may not accurately reflect the environment of the small intestine, a concern that has been previously noted. The study's focus on evaluating a device that could potentially overcome these limitations is both timely and compelling.

However, it is important to consider the practical challenges associated with the device, particularly in terms of sample recovery. The current method, involving a transporter collecting stool samples from participants' homes, could impose significant stress on participants. Additionally, if this device were to be implemented in a clinical setting, the associated human costs might limit its widespread adoption. Addressing these logistical challenges will be crucial for the device's practical application and acceptance in clinical practice.

The manuscript would benefit from a more detailed discussion on potential operational issues and troubleshooting strategies related to the use of the device. This includes, but is not limited to, device malfunctions, sample contamination, and data inconsistencies. Providing a comprehensive overview of these challenges, along with their respective mitigation strategies, would significantly strengthen the study's design and demonstrate its robustness to the readers.

Incorporating these considerations into the manuscript would not only enhance its academic depth but also provide a more realistic view of the device's potential application in clinical settings.

Author Response

Answer to reviewer 2:

We thank the reviewer 2 for his/her questions and suggestions. To clarify certain points, we added a new paragraph in the last part of the manuscript (lines 363 – 367): “Pelican Health is proactively addressing the burden of faeces collection imposed on volunteers in the current study by ensuring that pick-ups align with the volunteers' way of life in this study. Currently, we are developing approaches to facilitate both, the recovery of faeces and the medical device recovery from faeces, thereby lowering additional costs and patient burden.”

We understand the importance of minimizing any inconvenience to participants and are committed to making the process as seamless as possible.

(lines 368-371) “Sample contamination is a common issue frequently observed during employment of other non-invasive sampling tools [46] (“ J. Menard, S. Bagheri, S. Menon, Y. T. Yu, and L. B. Goodman, ‘Noninvasive sampling of the small intestinal chyme for microbiome, metabolome and antimicrobial resistance genes in dogs, a proof of concept’, Anim. Microbiome, vol. 5, no. 1, Art. no. 1, Dec. 2023, doi: 10.1186/s42523-023-00286-0”), and even during catheter aspiration [18]. However, the design and technical constraints incorporated into the development of the Pelican Health sampling device are specifically intended to limit these situations.”

Recognizing the common occurrence of data inconsistencies in microbiome research, we actively collaborate with the community to follow and provide input on guidelines that effectively mitigate these discrepancies and will guide our data analysis. Additionally, we acknowledge the need for more in-vivo trials to observe inconsistencies using the corresponding stool microbiome data as a reference, we are therefor committed to undertaking further research to address this aspect comprehensively.

Reviewer 3 Report

Comments and Suggestions for Authors

An important study to establish a novel technique allowing sampling of samples for assessment of microbiome in the small intestine, which has been challenging. Could the authors comment more on the specific regions of small intestinal samples to be collected by the device and the mechanisms of collection?

Author Response

Answer to reviewer 3:

We thank the reviewer 3 for his/her questions and suggestions. The medical device is composed of a Gastro-Resistant HPMC-Based “Next Generation Enteric” pill. This pill is produced and commercialized by Lonza (Switzerland). In 2022, they published a study in which they clinically proved the intestinal disintegration location of the pill (DOI: https://doi.org/10.3390/pharmaceutics14101999 ). In this study, 8 healthy volunteers ingested one Lonza Capsugel Next Generation Enteric capsule filled with 25 mg 13C-caffeine, 13.3 mg black iron oxide, 35 mg croscarmellose, and 216.7 mg standard capsule filling powder consisting of 99.5% mannitol and 0.5% silicon dioxide. Abdominal images (MRI images) were taken before the capsule intake, (t = −2 min), and at the following time points: 15 min, 30 min, 45 min, 60 min, 75 min, 90 min, 105 min, 120 min, 135 min, 150 min, 165 min, 180 min, 195 min, 210 min, 225 min, and 240 min. The aim is to evaluate the 13C3-labeled caffeine Pharmacokinetics (using a LC-MS/MS method). As results, they showed that none of the capsules showed any sign of disintegration in the stomach. According to the MRI, the capsules disintegrated in three subjects in the jejunum and in four subjects in the ileum. In one subject, the capsule reached the ascending colon within 45 min after ingestion.

In the manuscript, we added lines 129-130, 131-132 to clarify this point: “This capsule is produced by Lonza Capsules & Health Ingredients, (Colmar, France) and targets the distal parts of the small intestine [40]” (“[40] A. Rump et al., ‘In Vivo Evaluation of a Gastro-Resistant HPMC-Based “Next Generation Enteric” Capsule’, Pharmaceutics, vol. 14, no. 10, p. 1999, Sep. 2022, doi: 10.3390/pharmaceutics14101999.”…; lines 131-132: “The outer wall (capsule) then degrades once it reaches the distal jejunum/proximal ileum.”
